# Large electromagnetic field enhancement in plasmonic nanoellipse for tunable spaser based applications

Saqib Jamil[1]*, Waqas Farooq[1], Najeeb Ullah[2], Adnan Daud Khan[3], Usman Khan Khalil[4], Amir Mosavi[5,6,7]*

1 Department of Electrical Engineering, Sarhad University of Science and Information Technology, Peshawar, Pakistan, 2 University of Engineering & Applied Sciences, Peshawar, Pakistan, 3 US-Pak Center for Advanced Studies in Energy, UET Peshawar, Peshawar, Pakistan, 4 Department of Electrical Engineering, Cecos University, Peshawar, Pakistan, 5 John von Neumann Faculty of Informatics, Obuda University, Budapest, Hungary, 6 Institute of Information Society, University of Public Service, Budapest, Hungary, 7 Institute of Informational Engineering, Automation and Mathematics, Slovak University of Technology in Bratislava, Bratislava, Slovakia

* engr.saqibjamil@gmail.com (SJ); amir.mosavi@kvk.uni-obuda.hu (AM)

**Data Availability Statement:** All relevant data are within the paper and its Supporting Information files.

## Abstract

We theoretically demonstrated a class of plasmonic coupled elliptical nanostructure for achieving a spaser or a nanolaser with high intensity. The plasmonic ellipse is made up of gold film substrate. The proposed structure is then trialed for various light polarizations, moreover, a simple elliptical nanostructure has been chosen primarily from which different cases have been formed by geometry alteration. The structure supports strong coupled resonance mode i.e. localized surface plasmon (LSP). The localized surface plasmon resonance (LSPR) of the investigated system is numerically examined using the finite-element method (FEM). The calculations showed that the LSPR peaks and the local field intensity or near field enhancement (NFE) of the active nanosystem can be amplified to higher values by introducing symmetry-breaking techniques in the proposed ellipse and its variants. The coupled nanostructure having different stages of wavelengths can be excited with different plasmonic resonance modes by the selection of suitable gain media. In addition, a small-sized nanolaser with high tunability range can be developed using this nanostructure. The spaser phenomena are achieved at several wavelengths by changing light polarization and structure alteration methods. Giant localized field enhancement and high LSPR values enable the proposed model to be highly appealing for sensing applications, surface-enhanced Raman spectroscopy, and much more.

## Introduction

Lasers play an important role in physics and optics due to their coherent light-sourced nature. However, with increasing speed and size reduction in photonic devices, the role of traditional lasers is challenged due to the diffraction limit of light that usually prevents the miniaturization

**Funding:** The author(s) received no specific funding for this work.

**Competing interests:** The authors have declared that no competing interests exist.

of such devices less than half of its wavelength [1, 2]. The localized hotspot formation in nano-photonics with intensive field concentrations in plasmonic nanostructures is able to break the diffraction limit. The surface plasmon amplification by stimulated emission of radiation (SPA-SER) was first proposed by [3]. The plasmons behave like photons in conventional lasers with a plasmonic cavity which helps spaser to break the diffraction limit [4, 5]. The near field enhancement produced by nanostructures have a vast range of applications such as sensing [6], perfect light absorption [7], switching and nanocircuits [8], light speed manipulation and dispersion [9], cloaking and imaging [10], focusing and lasing [11], enhanced non-linear effect [12], surface-enhanced Raman scattering [13] and holograms. Coherent active plasmon-based structures open paths for the expansion of SP-based applications and investigation of the matter-light interactions at the nanoscale. Nanoscale field confinement and strong optical feedback are two critical factors for designing plasmonic lasers. The near field enhancement due to confinement can be realized by the excitement of plasmonic resonances in single metallic nanoparticles, dimer arrays and hybridized composite nanostructures [14]. LSPR modes can be controlled by the adjustment of nanoparticle's size and geometric configurations (shapes), while that of SPP modes are mainly controlled by the variation in the array's period or incident angle. Theory and experiments have shown that a metal nanoparticle array supports the SPP and LSPR coupling, where the 2-D array of the metallic nanoparticles help as a grating coupler to excite the SPP mode on the nearby metallic film [15]. The LSPR systems with strong near-field enhancement are useful for SERS and surface enhanced coherent anti-Stokes Raman scattering (CARS) [16], optical based sensors and absorption. In [17], the authors have proposed a sensor design based on metal-insulator-metal (MIM) configuration. The investigated model can be deployed for temperature and biosensing applications. The model is made up from two resonant cavities having circular and square shape, with an MIM waveguide coupled to one side. For operating the device in biosensing mode, the analytes were injected through square cavity while the thermo-optic polymer was deposited on the circular cavity, which provided a shift in resonance wavelength in accordance with ambient temperature. The proposed design is much beneficial for testing of biological analytes in controlled temperature environment and for reduction the fluctuation of ambient temperature on refractometric measurements. A highly sensitive plasmonic filter having centrally coupled ring resonator was numerically investigated by [18]. The suggested model was able to operate as a plasmonic index sensor with sensitivity and figure of merit (FOM). The silver nanorod in the resonator provided tuneability to improve sensitivity and FOM. The central coupling showed high FOM and sensitivity compared to the side coupling method due to which the peaks can be realized for a vast wavelength range enabling the method to be useful for optical communications and on-chip plasmonic sensing. The authors in [19], have proposed a plasmonic perfect absorber (PPA) having an array of metal nanorod with connected veins for ultra-sensitive sensing of refractive index in the near-infrared region. A cavity resonance center can be constituted with PPA which served as a plasmonic sensor. The vein effect has a dominant influence on plasmon resonance resulted in a high-quality factor and figure of merit. The fabrication of metal-dielectric nanorod array is done by [20], by using nanosphere lithography with ion etching method. The surface and gap plasmons were numerically demonstrated by using the finite element method. The proposed method provided a practical sensing platform with a cost-effective fabrication method. In [21], the authors have used the laser-direct writing technique to fabricate Ag nanostructures for SERS applications. Multi-level Raman imaging of organic molecules was observed by controlling laser powers. The phenomena was further investigated by using atomic-force microscopy and electromagnetic calculations. The method provided a promising development in low-cost sensing chip.

We investigated the coupling of plasmons in a simple elliptical-shaped nanostructure and its variants made up from silica and truncated gold ellipse, which can be fabricated using

techniques suggested by [22, 23]. The distinctive feature of an elliptical nanostructure is that symmetry breaking can be simply obtained by extracting a small portion from the main ellipse and by rotating them at certain angles results in some subdivisions or varieties. This cannot be obtained with spherical symmetries or circular-based nanostructures. We started our study from a single nano-ellipse and moved towards dimer, trimer, quadramer, and their possible configurable arrangements. Near-field enhancements are accomplished at different wavelengths by producing variations in the structure and incident polarizations. Different cases have been studied along with x-y polarization and the effect of the incoming field on output spectra with local field enhancement. Furthermore, symmetry breaking has been done in two steps. First by taking a part from the full ellipse and then giving a particular angular shift to these parts. The extinction spectrums of all the configurations are strongly dependent on the polarization of the incident light. Eventually, the optical response results of ellipse and variants are studied briefly and their outcomes are well explained. We studied the extinction spectrum, distribution of the induced surface charges, and the near field enhancements of gold-silica elliptical nanostructure and its variants. At the same time, this configuration has rounded corners due to which it can be fabricated using imprinting lithography, atomic force microscopy, or the methods explained by [20, 21].

## Model

In this work, we have investigated several patterns of single nano-ellipse (NE) such as nano-elliptical dimer (NED), linear chain nano-elliptical trimer (LCNET), linear chain nano-elliptical quadramer (LCNEQ), symmetry broken nano-ellipse (SBNE), symmetry broken nano-elliptical dimer (SBNED), symmetry broken linear chain nano-elliptical trimer (SBLCNET) and symmetry broken linear chain nano-elliptical quadramer (SBLCNEQ). All these configurations have been formed from a single NE by adding an identical NE or by extracting a portion and then forming above mentioned configurations which are briefly discussed in later sections. The transformation from single NE to various sets is shown in Fig 1. We found that such arrangements are much valuable for spectral lines and near field enhancement (NFE). Furthermore, we performed a polarization-based simulation in which we have extracted results for x-y polarizations along with geometric alterations for bringing a vast range of wavelength range along with giant NFE for spaser-based applications such as lithography,

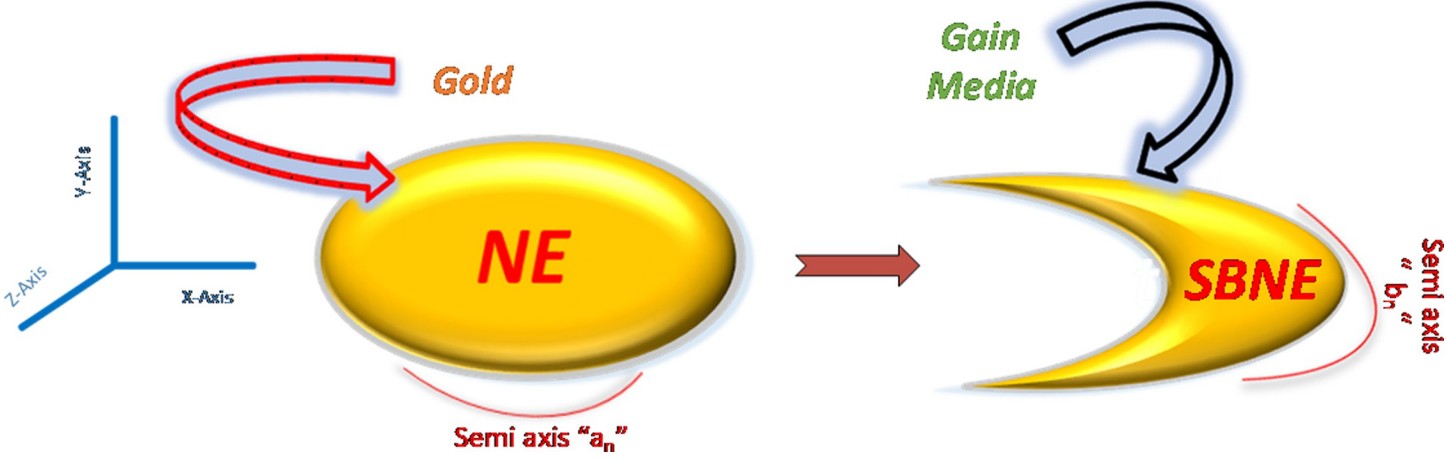

**Fig 1. The transformation from single nano-ellipse (NE) to symmetry broken nano-ellipse (SBNE).**

switching, imaging, and SERS, etc. COMSOL with RF module has been utilized for finding optical properties/outcomes of the proposed nanostructure and its variants. Johnson and Christy's data is used to find out the dielectric constant of gold [24] with air as a surrounding medium. For simplicity, the value of gain media is set as 2.42 for all the cases. The proposed model is placed in a Cartesian coordinate system in which light is incident from x-direction and y-direction respectively. The dimensions of the nano-structure are set as '$a_n$' and '$b_n$' where 'n' is an integer (1,2,3,4) for various configurations. The NE is transformed to a phase known as the symmetry breaking phase and the term used for nano-ellipse is symmetry broken nano-ellipse SBNE. From this set of designs, we have modeled different patterns mentioned above.

## Results and discussion

In this section, we have briefly described all the cases for which we have calculated our results and performed the simulations. For a detailed understanding/analysis of the optical properties of a gold–silica elliptical nanostructure, it is necessary to study a gold-silica nanostructure. The optical outcome of an ellipsoid is investigated by using the plasmon hybridization theory [25].

### A. Optical properties of a nano-ellipse (NE)

We consider the truncated gold nano-ellipse (NE) surrounded by a gain media shown in Fig 2A. The parameters of the NE are set as $a/b/t = 50/30/25\ nm$ respectively. Where $a$ is the outer semi-axis $b$ represents the inner semi-axis while $t$ gives the thickness of the NE. The structure is placed along the x-axis, its response is calculated for the x-polarization case and y-polarization case. That is, the light is the first incident from the x-axis and in the second case, the position of the nano-ellipse remained unchanged while the light was incident from the y-axis. We calculated the extinction spectra in Fig 2B. and amplification obtained by the light interaction with the nanostructure. The extinction spectrum is affected by changing the direction of light on the NE, since for x-polaroid, the spectrum is red-shifted with the peak occurred at *915 nm* due to transverse dipolar mode and corroboration by the charge distribution, while for the y-polaroid case the spectra show a blue shift with a minor peak obtained at *606.2 nm* since the

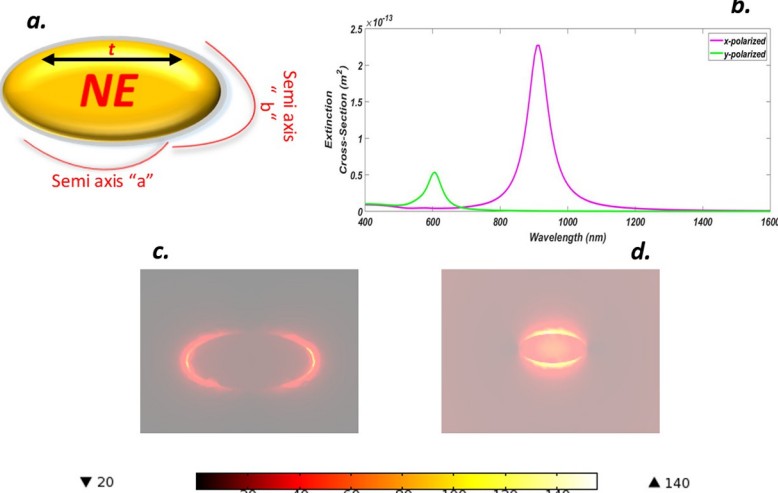

**Fig 2.** (a) Truncated Gold Nano-ellipse (NE) with gain media coating (b) ECS of NE for x-y polarization (c) NFE of NE for x-polarization d. NFE of NE for y-polarization.

transverse dipolar mode is weakened. The lowest energy mode corresponds to an anti-symmetric coupling between the plasmon resonance and the metallic nanostructure which in this case is shown by the y-polaroid case. While high peak arises from symmetric coupling between light and the NE. All of the systems studied herein were assumed to be in a vacuum. Similarly, amplification of both cases can be seen in Fig 2C and 2D. It is clear that efficient coupling results in plasmon hybridization which in turn develops plasmonic effect due to which "hot-spots" are formed, so strong antibonding modes were formed for the x-polarization case since, the light was efficiently coupled with the NE that led to an amplification factor of about *126* and for y-direction, light is poorly coupled resulting the formation of bonding modes, hence, the strong plasmonic effect does not occur and we obtained less amplification of *18*. The local field enhancement (LFE) for the they-polarized case was *7* times less than that of the x-polarized case but can still be used for applications falling in this frequency range.

## B. Optical properties of a symmetry broken nano-ellipse (SBNE)

Metal-based nanoparticles support plasmon resonance(s) whose energies are strongly dependent on the geometry of the nanostructure. The resonance tunability feature has brought considerable experimental and theoretical research [26]. An important parameter in plasmonics is the effect of symmetry breaking and for nanostructures much smaller than the wavelength of the incident light, only plasmons with finite dipole moments can be excited. Nanoshells are highly symmetric nanostructures in which symmetry breaking technique is normally introduced by displacing a core concerning the outer shell may activate high-order mode(s) that become visible in the optical spectrum [27]. We have now considered a symmetry breaking case in which we have scratched a portion from the main ellipse by setting *a/b/t = 50/30/25 nm* respectively and this case is termed as symmetry broken nano-ellipse (SBNE). The SBNE is surrounded by a gain media to compensate against ohmic losses as shown in Fig 3A. We have shown that symmetry breaking leads to much larger field enhancement and the higher multi-polar modes become dipole active through hybridization. Dark quadrupolar plasmon starts coupling with superradiant antibonding dipolar mode which induces an asymmetric Fano resonance in the extinction spectra. By simply changing the polarization the incident light can be directly coupled to quadrupolar mode. This coupling interferes with the dispersive coupling

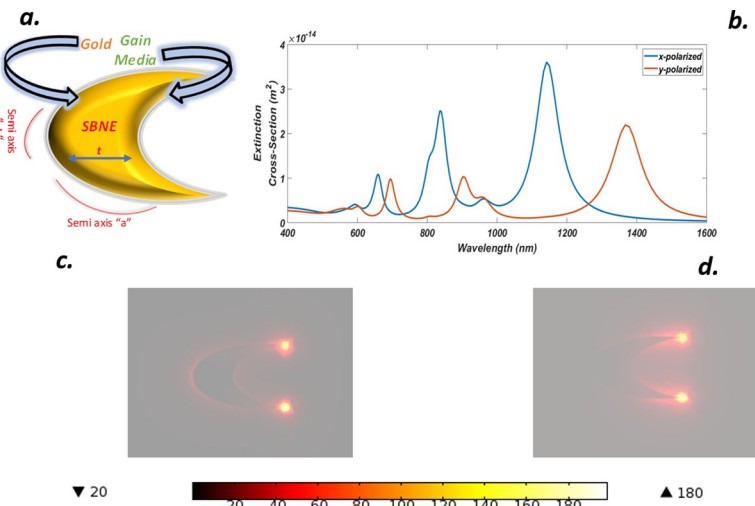

**Fig 3.** (a) Truncated Gold Symmetry broken nano-ellipse (SBNE) with gain media coating (b) ECS of SBNE for x-y luminance (c) NFE of SBNE for x-polarization (d) NFE of SBNE for y-polarization.

between the superradiant and quadrupolar mode and alters the line shape of the Fano resonance. We have calculated the extinction cross-section (ECS), for x-y polarization which shows multiple peaks as shown in Fig 3B. For x-polarization, the highest peak occurred at *1142 nm* and shows a redshift compared to the other two peaks that occurred at *836.4 nm* and *659.9 nm*. The peaks at the wavelengths of *836.4 nm* and *659.9 nm* shows low energy and are blue-shifted since the alignment of dipolar modes was antisymmetric. Similarly, if we look at the extinction spectra for y-polarization in the same plot we see that the strongest peak occurred at *1367 nm* and shows red shifting compared to x-polarization as well as for the they-polarized case, but still, it can be noticed that its height is much low as compared to x-polaroid due to antisymmetric interaction of light with the SBNE. Similarly, the other two peaks were appeared at *902.6 nm* and *694.3 nm* and were blue-shifted concerning y-polaroid but show a slight red shifting concerning x-polaroid peaks. Also, it is noticeable that symmetry breaking led to multi-wavelength operation compared to. Fig 3C and 3D shows the near field enhancements (NFE) for the x-y polaroid SBNE case. The electric field distribution near the surface shows the typical dipolar resonance properties. The large electric field occurs for the x-polarized case whose value goes up to *200*, and for y-polarization, the enhancement value was recorded to about *118*. Furthermore, amplification obtained from the x-polarized case is about *1.5* times the y-polarized case as well as the full elliptical nano-structure. It can be seen that the y-polarized case for SBNE configuration produced far better results as compared to NE, whose value was 18 which is 6.5 times less than the current case. Hence, symmetry breaking produced huge near-field enhancement (NFE) values along with x-y polarization with multiple peaks.

## C. Optical properties of a nano-elliptical dimer (NED)

Plasmonic nano-elliptical dimer (NED) with closed spaced resonant particles is an arrangement in which two identical ellipses are brought close to each other and the arrangement is termed as nano-elliptical dimer (NED) structure. The coupling between two NE structures with a small separation can induce a strong, enhanced, and deep subwavelength-confined optical near field inside the narrow gap by incident light. In Fig 4A such arrangement is shown with an outer semi-axis a = 90 nm and an inner semi-axis b = 60 nm. The gap between two nano-structures was set as s = 5nm, t = 40 nm for both NE structures and the light was x-y

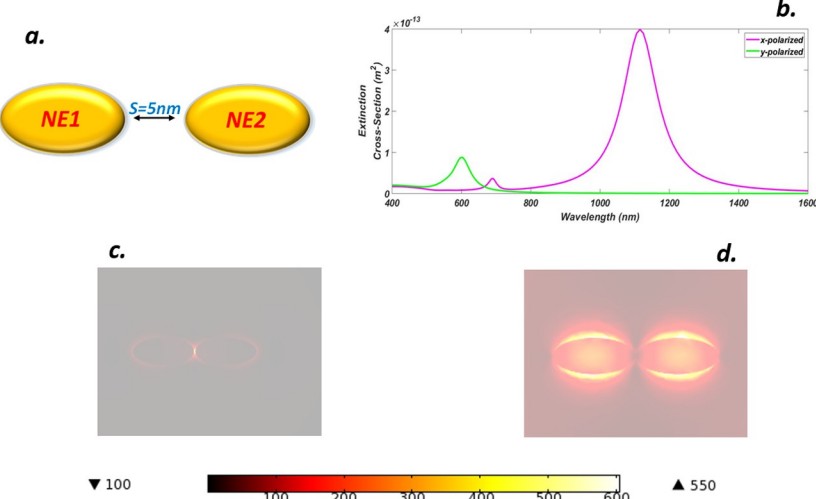

**Fig 4.** (a) Truncated Gold nano-elliptical dimer (NED) with gain media coating and separation S = 5 nm (b) ECS of NED for x-y luminance (c) NFE of NED for x-polarization (d) NFE of NED for y-polarization.

sparkled. We have plotted the extinction cross-section of the NED Fig 4B, which exhibits a typical pattern when excited with x-polarized light (i. e. all-electric field components are parallel to the long axis of dimer). A large red-shifted peak can be seen at *1115 nm* which typically represents the strong coupling of the dimer with an incoming field. Another peak appears at the wavelength of *691.7 nm* for the same polarization but its magnitude is too small. A blue-shifted peak shown in green color presents a y-polaroid case and this shifting along with the small height of the peak depicts subradiant quadrupolar nature resulting due to poor interaction of the electric field with the NED at *600 nm*. Fig 4C shows the formation of the hot spot and amplification obtained along the x-axis. The charges are uniformly distributed around the dimer but the hot spot is formed between the gap and this is a maximum intensified point for which the enhancement maxima reaches above *620* which shows a very high amplification factor for the arrangement. Similarly, Fig 4D shows the local field factor for the y-polaroid case and the picture clearly shows the arrangement of charges around the dimer. In this arrangement, no clear hot spot is visible due to subradiant modes, This shows when light is incident from the y-axis efficient coupling is not possible which results in the amplification factor of *13* about which is about 48 times smaller than the x-polaroid case.

## D. Optical properties of a symmetry broken nano-elliptical dimer (SBNED)

In this section, we have engineered a symmetry broken nano-elliptical dimer SBNED by setting all the structural parameters the same as that of section B and adding SBNE as depicted in Fig 5A. The SBNED is an ideal candidate for analysis since it supports the polarization of light at different angles, multipolar progression, and dark modes whose tunability can be adjusted by incident light. SBNED instigated dark plasmonic modes to overlap with a bright mode resulting in the fortification of the plasmonic effect. The extinction spectra for x-y luminance are shown in Fig 5B. Five and three peaks appeared for x-polarization and y-polarization respectively because of the angular momentum contributions. The peak formed at the wavelength value of *1494 nm* (blue-line) shows red shifting and represents the negative parity dipoles.

However, heights are shown at *1199 nm* and *865.4 nm* represent strong mixing of positive parity bright modes with dark modes. The peak at the wavelength of *680.6 nm* shows blue

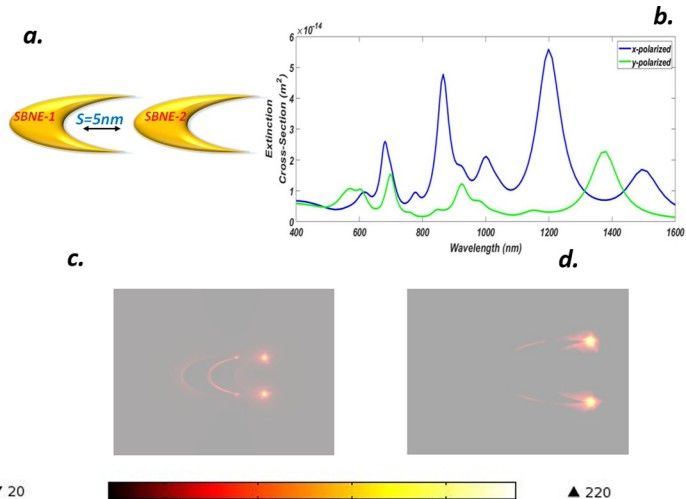

**Fig 5.** (a) Truncated Gold symmetry broken nano-elliptical dimer (SBNED) with gain media coating and separation S = 5 nm (b) ECS of SBNED for x-y luminance (c) NFE of SBNED for x-polarization (d) NFE of SBNED for y-polarization.

shifting with a strong magnitude compared to the *1001 nm* peak. Similarly, y-polarization produced a peak at *1381 nm*, showing a dipolar mixture of both NEs. Dark quadrupolar modes created low magnitude peaks at the wavelengths of *921.6 nm* and *699.3 nm* respectively. The tuning in these plots may be developed by alteration in the geometry or gap variation which is a highly suitable biosensor [28]. Fig 5C and 5D shows the NFEs for x-y polarizations, in which the distribution of charges for x-luminance is far better than y-luminance. The charge distribution occurs in the gaps along with the outer surfaces of each NE. Four hot spots are visible, two are between the gap position and the other two at the tips of the NE showing that a huge plasmonic effect occurred at these points, and charges are strongly confined and produced an NFE value of 238. Similarly, y-polarization shows less charge distribution around the structure presenting weak hybridization and producing NFE of about 110.

### E. Optical properties of a linear chain nanoelliptical trimer (LCNET)

When a plasmonic nanoparticle is built from more than two nanoparticles, both the shape and energy of its plasmonic modes strongly depend on the arrangement of the individual particle. We now extend our work to three elliptical parts i.e each NE is placed along an x-y plane with a fixed separation of *s = 5 nm* forming a linear chain nano-elliptical trimer (LCNET) configuration in which each NE is covered with the gain media layer, the LCNET is tailored in such a way that both semi-axis value and thickness for each NE remained unchanged and kept fixed at *a = 50 nm* and *b = 30 nm* and *t = 25 nm* respectively as shown in Fig 6A. Trimer configuration offers a rich spectrum of possible outcomes for coupling between nanoparticles. New physical phenomena may be realized by symmetry alterations. The extension is performed to further optimize the results and bring a variety of wavelength ranges for multiple applications. Fig 6B shows the extinction spectrum in which strong bright dipole-dipole bonding produced a peak for an x-polaroid case at *902.1 nm*, while at *684.3 nm* another peak appeared due to dark octupolar modes. A Fano resonance is visible at about *568 nm*. Similarly, for the y-polaroid case, a weak and low height peak occurred at *537 nm* due to octupolar modes since for this case the light coupling with the structure was not efficient hence failed to generate antibonding modes. Fig 6C and 6D shows the distribution of charges and formation of spots for each x-y polarization respectively. It can be seen that strong hot spots are formed in the gap positions in

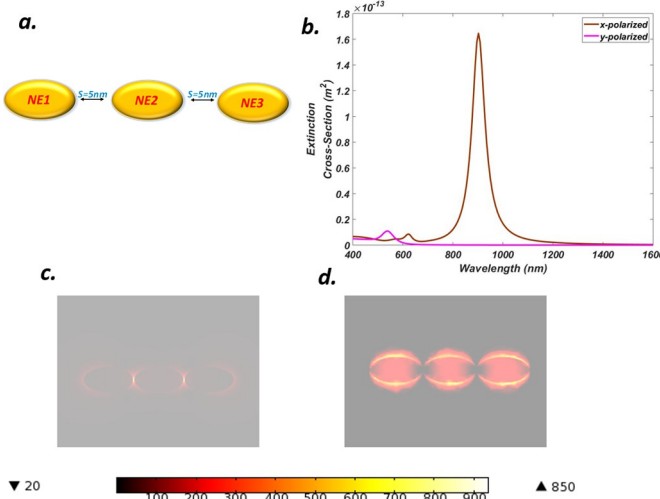

**Fig 6.** (a) Truncated Gold linear chain nano-elliptica trimer (LCNET) with gain media coating and separation S = 5 nm (b) ECS of LCNET for x-y luminance (c) NFE of LCNET for x-polarization (d) NFE of LCNET for y-polarization.

Fig 6C due to efficient coupling while in Fig 6D, most of the plasmons are scattered and fail to be confined between structural gaps, hence a large NFE of *930* is obtained for the x-polaroid case while this value dropped to about *7* for the y-polaroid case which is about *132* times less than x-luminance version. Furthermore, the x-polaroid NFE value is much high compared to all the preceding cases while y-polaroid is the smallest. Similarly, the energy level of the peak is also low compared to the above cases.

## F. Optical properties of a symmetry broken linear chain nano-elliptical trimer (SBLCNET)

We have now brought three symmetry broken elliptical parts i.e each SBNE is placed in the x-y plane having the same parameters as that of section B, thus forming a symmetry broken linear chain nanoelliptical trimer (SBLCNET) with a fixed separation of *s = 5 nm* as shown in Fig 7A. Fig 7B shows the extinction spectrum of the SBLCNET in which it can be seen that for each case i.e x-y polarization four peaks have appeared. The blue line presents x-polarization and it offered a wavelength spectrum in the range of *600 nm* to *1650 nm* approximately, with the highest rise at *1222 nm* formed due to the strong mixing of bright modes. Another broad red-shifted peak is visible at *1587 nm* which represents antibonding modes. Two blue-shifted peaks were raised at the wavelengths of *865.4 nm* and *711 nm* respectively, with the formation of Fano resonances. Similarly, SBLCNET offered multiple wavelengths ranges with slightly low heightened peaks for y-polarization at the positions of *1366 nm*, *928.3 nm*, *699.3 nm*, and *573.5 nm* respectively. Hence this configuration generated a multi-wavelength spectrum range for numerous optical applications covering this area. The near field enhancement result for x-polarization is shown in Fig 7C in which the charges have captured all the gaps between each unit but the major hot spots are visible at the two tips of the SBNE-3. This occurred due to the strong coupling of light and LCNET which energized the dark and bright modes through hybridization and produced a NFE of about *605*. Similarly, the y-polarization case is depicted in Fig 7D in which the charges are distributed between the gaps in the LCNET structure. However, strongly confined hot spots were not generated in this case due to poor coupling of light and formation of dark modes but still, a weak hot spot version at the tips of the SBNE-3 may

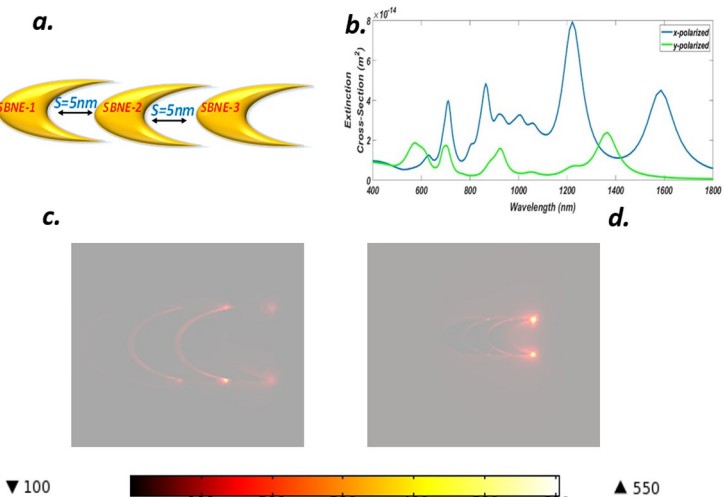

**Fig 7.** (a) Truncated Gold symmetry broken linear chain nanoelliptical trimer (SBLCNET) with gain media coating and separation S = 5 nm (b) ECS of SBLCNET for x-y luminance (c) NFE of SBLCNET for xpolarization (d) NFE of SBLCNET for y-polarization.

be found which shows that the majority of charges were confined at these points and led to the near field enhancement of about *128*. The NFE produced by y-polaroid luminance is far more than compared to the y-polaroid of LCNET while the x-polaroid NFE value of SBLCNET is less compared to LCNET, however, symmetry breaking produced a vast range of wavelength spectrum which in turn introduces tunability and may be deployed for the number of applications.

### G. Optical properties of a linear chain nanoelliptical quadramer (LCNEQ)

We further extend our investigation by adding another NE with a previous LCNET structure, introducing three gaps between each NE, and again the separation is kept constant at *s = 5 nm*. Also, the semiaxis parameters are fixed at *a = 50 nm* and *b = 30 nm* with thickness *t = 25 nm* respectively for each NE, the overall arrangement is termed as linear chain nanoelliptical quadramer (LCNEQ). The angle between each SBNE is kept as θ = 0˚ along with the gain media coating as shown in Fig 8A. The extinction spectrum is shown in Fig 8B and is identical to Fig 6B in shape with a peak that appeared at *956.2 nm* due to bright dipole-dipole bonding modes but has high energy. Similarly, dark octupolar modes activated at *684.3 nm*, and a Fano resonance was achieved at *597.3 nm*. The y-polaroid curve is also identical to Fig 6B with a small peak that appeared due to octupolar modes at *535 nm*. The charge distributions and hot spot formation for x-y luminance is shown in Fig 8C and 8D respectively with a NFE value of about 1019 and 8 for each case.

### H. Optical properties of a symmetry broken linear chain nano-elliptical quadramer (SBLCNEQ)

Here we have tailored SBLCNET by adding another SBNE to form symmetry broken linear chain nanoelliptical quadramer (SBLCNEQ) as shown in Fig 9A. This configuration has been developed for further investigating the effects of polarization. The extinction spectrum is shown in Fig 9B, which offers a rich wavelength range by producing five peaks for the x-polaroid case with healthy heights. The peak at *1627 nm* and *1234 nm* is the highest peak

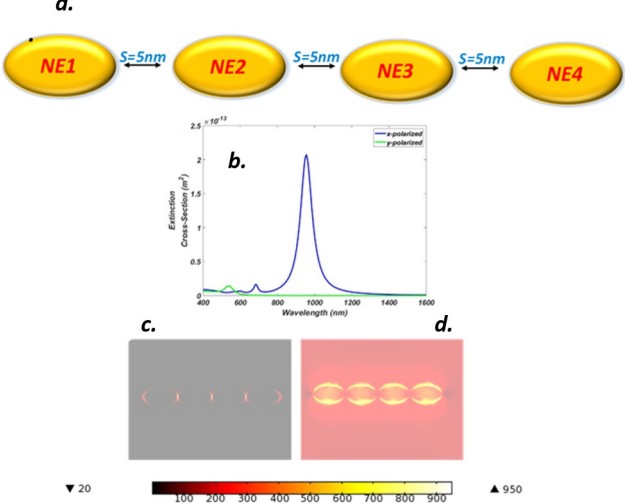

**Fig 8.** (a) Truncated Gold linear chain nano-elliptical quadramer (LCNEQ) with gain media coating and separation S = 5 nm (b) ECS of LCNEQ for x-y luminance (c) NFE of LCNEQ for x-polarization (d) NFE of LCNEQ for y-polarization.

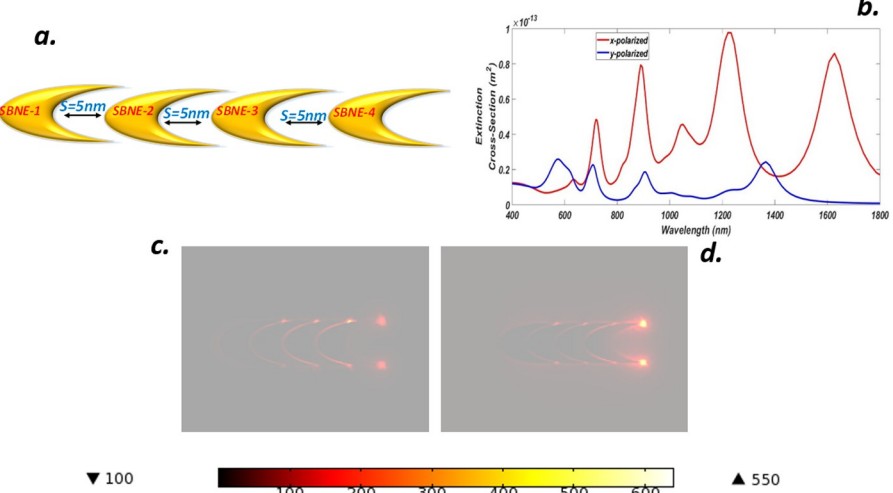

**Fig 9.** (a) Truncated Gold symmetry broken linear chain nano-elliptical quadramer (SBLCNEQ) with gain media coating and separation S = 5 nm (b) ECS of SBLCNEQ for x-y luminance (c) NFE of SBLCNEQ for x-polarization (d) NFE of SBLCNEQ for y-polarization.

representing the efficient coupling of light and activation dipole-dipole modes. A small peak appeared at *1051 nm* due to negative parity mode. Furthermore, a peak at *889.5 nm* shows a good hybridization of dark and bright modes while a *719 nm* peak represents a blue-shifted peak and has a relatively small height compared to other peaks. Similarly, the y-polaroid case is shown by a blue line with four peaks at *1366 nm*, *908.5 nm*, *707 nm*, and *573.5 nm* respectively. This case also provided a good wavelength range but the peak heights are far low compared to the x-polarized version mainly due to poor interaction of incident light from y-direction causing less mixing of each SBNE charge. Fig 9C and 9D depicts the near field enhancement (NFE) for x-y polarizations. Since charge confinement mainly occurs between the gaps and SBLCNEQ offered such gap ranges hence the enhancement achieved for x-polarized reaches up to *640*. Similarly, for y-polarization, the light was poor coupled with the SBLCNEQ compared to x-direction hence the enhancement value recorded for this case is about *124*. Hence, this configuration offers a vast wavelength range but a slight low NFE values for either case, compared to LCNEQ.

Finally, we have summarized all the targeted parameters for x-y polarization that are achieved so far from this study in Tables 1 and 2 respectively. The details listed in both tables

**Table 1. Characteristic parameters of the elliptical nano-structures for the x-polarization with outer semi-axis a = 90 nm and inner semi axis b = 60 nm, thickness t = 40 nm for NE, NED, LCNET and LCNEQ.**

| Structure Type | NE | SBNE | NED | SBNED | LCNET | SBLCNET | LCNEQ | SBLCNEQ |
|---|---|---|---|---|---|---|---|---|
| No. of Peaks | 1 | 3 | 2 | 5 | 3 | 4 | 3 | 5 |
| Wavelength (nm) | 915 | 659.9 | 691.7 | 680.6 | 568 | 711 | 597.3 | 719 |
| | | 836.4 | 1115 | 865.4 | 684.3 | 865.4 | 684.3 | 889.5 |
| | | 1142 | | 1001 | 902.1 | 1222 | 956.2 | 1051 |
| | | | | 1199 | | 1587 | | 1234 |
| | | | | 1494 | | | | 1627 |
| NFE | 126 | 200 | 620 | 238 | 930 | 605 | 1019 | 640 |

While fixing a = 50 nm and b = 30 nm, thickness t = 25 nm for SBNE, SBNED, SBLCNET and SBLCNEQ.

**Table 2. Characteristic parameters of the elliptical nano-structures for the y-polarization with outer semi-axis a = 90 nm and inner semi-axis b = 60 nm, thickness t = 40 nm for NE, NED, LCNET, and LCNEQ.**

| Structure Type | NE | SBNE | NED | SBNED | LCNET | SBLCNET | LCNEQ | SBLCNEQ |
|---|---|---|---|---|---|---|---|---|
| No. of Peaks | 1 | 3 | 1 | 3 | 1 | 4 | 1 | 4 |
| Wavelength (nm) | 606 | 1367 | 600 | 699.3 | 537 | 577.5 | 535 | 575 |
| | | 902.6 | | 921.6 | | 699.3 | | 707 |
| | | 694.3 | | 1381 | | 928.3 | | 908.5 |
| | | | | | | 1366 | | 1366 |
| NFE | 18 | 118 | 13 | 110 | 7 | 128 | 8 | 124 |

While fixing a = 50 nm and b = 30 nm, thickness t = 25 nm for SBNE, SBNED, SBLCNET, and SBLCNEQ.

further describe that symmetry breaking provides a variety of peaks and a vast area along the wavelength spectrum. These parameters depict the dominancy of x-polaroid cases over y-polaroid mainly due to light coupling effects. Table 3 lists the structure type/ names with abbreviations for convenience. Furthermore, Figs 10 and 11 show the achieved NFE for x-y polaroid cases in which it can be seen that simple structure configurations for x-polarization show higher values compared to y-polaroid. However, this is not the case for symmetry breaking in which high NFE values have been achieved for even y-polaroid cases, which shows that the conducted study is much useful for a variety of SPASER-based applications.

## Conclusions

The generation of high order plasmon mode(s) has been investigated along with large near field enhancement (NFE) in a nano-elliptical configuration and its variants for x-y polarized light. Simple NE showed a uniform distribution of charges and hot spot formation with a good line shape. Same parametric values were applied to a NED configuration which offered an efficient coupling of dark quadrupolar with bright dipolar modes which developed two peaks along the spectrum. Symmetry breaking technique has been applied on NE to form an SBNE structure, since symmetry breaking is an ideal testbed for enhancement layout of applied parameters. This method brought three peaks at different wavelengths due to the mixing of odd and even modes. We further performed extensions to form trimer and quadramer structures that brought a huge variety of wavelengths with the formation of multiple peaks and hotspots along with high NFE values. To, conclude mono, dimer, trimer and quadramer simple and symmetry broken variants are ideal for producing well-defined resonance tunability and higher-order dark plasmonic modes in the near-infrared (NIR) and visible regions, along with

**Table 3. Structure types and respective abbreviations.**

| Structure type/name | Abbreviation |
|---|---|
| NE | Nanoellipse |
| SBNE | Symmetry Broken Nano-Ellipse |
| NED | Nano-Elliptical Dimer |
| SBNED | Symmetry Broken Nano-Elliptical Dimer |
| LCNET | Linear Chain Nano-Elliptical Trimer |
| SBLCNET | Symmetry Broken Linear Chain Nano- Elliptical Trimer |
| LCNEQ | Linear Chain Nano-Elliptical Quadramer |
| SBLCNEQ | Symmetry Broken Linear Chain Nano-Elliptical Quadramer |

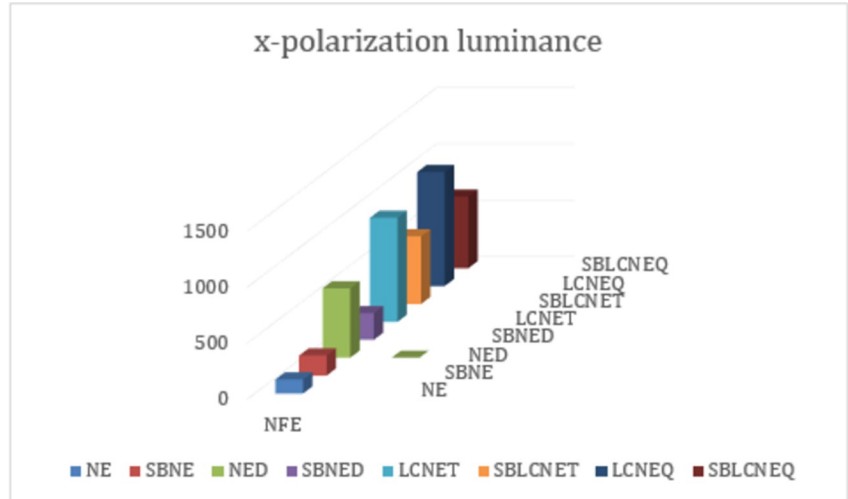

**Fig 10. Near field enhancement (NFE) of normal and symmetry broken nano-elliptical structures for x-polarization luminance.**

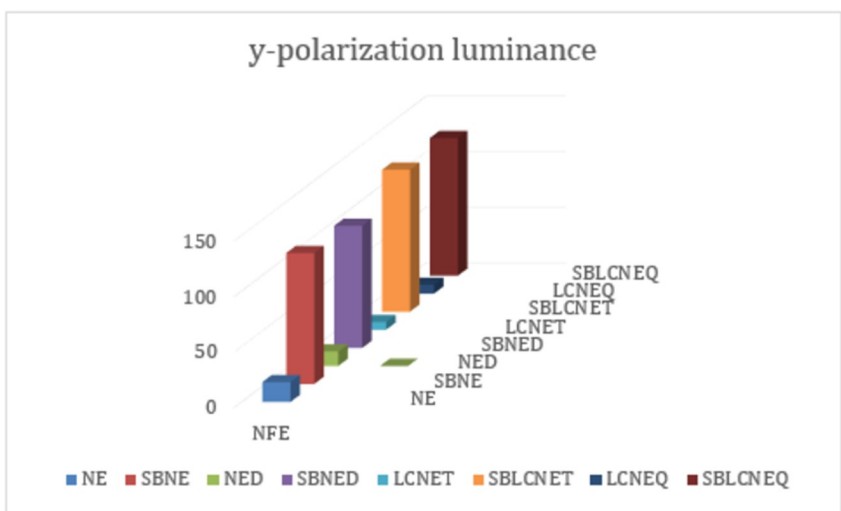

**Fig 11. Near field enhancement (NFE) of normal and symmetry broken nanoelliptical structures for y-polarization luminance.**

huge NFE values which may be useful for numerous spaser based applications like switching, SERS, slow light and bio-medical.

## Supporting information

**S1 Dataset.**
(PDF)

## Acknowledgments

The authors would like to thank Sarhad University of Science and Information technology, Peshawar, Pakistan and Obuda University, Budapest, Hungary. One of the authors would like

to thanks Dr. Javed Iqbal for his valuable and constructive suggestions during the planning and development of this research work.

## Author Contributions

**Conceptualization:** Saqib Jamil, Waqas Farooq, Adnan Daud Khan.

**Data curation:** Saqib Jamil, Waqas Farooq, Usman Khan Khalil.

**Formal analysis:** Waqas Farooq, Najeeb Ullah, Adnan Daud Khan, Usman Khan Khalil.

**Funding acquisition:** Amir Mosavi.

**Investigation:** Saqib Jamil.

**Methodology:** Saqib Jamil, Waqas Farooq, Adnan Daud Khan.

**Project administration:** Adnan Daud Khan.

**Resources:** Saqib Jamil, Waqas Farooq.

**Software:** Saqib Jamil, Adnan Daud Khan.

**Supervision:** Adnan Daud Khan.

**Validation:** Saqib Jamil, Adnan Daud Khan.

**Visualization:** Saqib Jamil, Waqas Farooq.

**Writing – original draft:** Saqib Jamil.

**Writing – review & editing:** Saqib Jamil, Waqas Farooq, Najeeb Ullah, Adnan Daud Khan, Usman Khan Khalil.

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
