## [Decision Letter · Decision Letter 0]

23 Dec 2021

PONE-D-21-35833Large Electromagnetic Field Enhancement in Plasmonic Nanoellipse for Tunable Spaser Based ApplicationsPLOS ONE

Dear Dr. Waqas Farooq,

Thank you for submitting your manuscript to PLOS ONE. After careful consideration, we feel that it has merit but does not fully meet PLOS ONE’s publication criteria as it currently stands. Therefore, we invite you to submit a revised version of the manuscript that addresses the points raised during the review process.

We look forward to receiving your revised manuscript.

Kind regards,

Yuan-Fong Chou Chau

Academic Editor

PLOS ONE

Journal Requirements:

4. Please include a copy of Table 2 which you refer to in your text on page 16.

Reviewers' comments:

Reviewer's Responses to Questions

**Comments to the Author**

1. Is the manuscript technically sound, and do the data support the conclusions?

Reviewer #1: Yes

Reviewer #2: Yes

2. Has the statistical analysis been performed appropriately and rigorously? 

Reviewer #1: Yes

Reviewer #2: Yes

3. Have the authors made all data underlying the findings in their manuscript fully available?

Reviewer #1: Yes

Reviewer #2: Yes

4. Is the manuscript presented in an intelligible fashion and written in standard English?

Reviewer #1: Yes

Reviewer #2: Yes

5. Review Comments to the Author

Reviewer #1: The authors theoretically demonstrated a class of plasmonic coupled elliptical nanostructure for achieving a spaser or a nanolaser with high intensity. Technically, the authors employ the numerical methods with apparent authority and the results seem valid. This work is interesting for the readership and deserve publication after major revision. This work will be more impactful if the authors address the following comments.

1. Figs. 2-9 (c) and (d) are blurred. Please replace them as the clear ones.

2. The references used in introduction section should be improved. It is written in abstract section that “Giant localized field enhancement and high LSPR values enable the proposed model to be highly appealing for sensing applications, surface enhanced Raman spectroscopy”. To be beneficial for the readers to enrich the mechanism and background of plasmonic sensor and surface enhanced Raman spectroscopy, the authors should mention the other approaches of plasmonic sensor, i.e., Nanomaterials, 10, 1399 (2021), J Phys D: Appl Phys, 54, 115301(2021)), Results in Physics, 15, 102567(2019) and Nanomaterials, 9,1691 (2019), and for surface enhanced Raman spectroscopy (i.e., Optics Express, 21, 24460 (2013)).

Reviewer #2: The authors reported a class of plasmonic coupled elliptical nanostructure for achieving a spaser or a nanolaser with high intensity. The FEM calculations showed that the LSPR peaks and the local field intensity or near field enhancement (NFE) of the active nanosystem can be amplified to higher values by introducing symmetry-breaking techniques in the proposed ellipse and its variants. Giant localized field enhancement and high LSPR values enable the proposed model to be highly appealing for sensing applications, surface enhanced Raman spectroscopy, and much more. The research is significant, the amount of data is large, and the contrast for characteristic parameters of different-type nano-structures is clear. However, before possible publication, some of the listed points should be explained and revised for further improving the manuscript.

1. Authors theoretically investigated a class of plasmonic coupled elliptical nanostructure for achieving spaser, but the shape of SBNE structures were unusual and whether they can be prepared for practical application?

2. Why the structure type with maximum NFE for Elliptical nano-structures for the x polarization was SBLCNET, but the structure type with maximum NFE for Elliptical nano-structures for the y polarization was LCNEQ rather than SBLCNET?

3. The calculated results should be supported by some experimental data through experimental measurement or experimental results from literatures.

4. How does the performances (such as NFE) of the structures in this work when compared with that in literatures.

5. There were too many abbreviations, so it is recommended to put them in a table.

6. The table note for Table 3 should be Table 2.

6. PLOS authors have the option to publish the peer review history of their article (what does this mean?). If published, this will include your full peer review and any attached files.

Reviewer #1: No

Reviewer #2: No

---

## [Author Response · Author response to Decision Letter 0]

18 Jan 2022

Response to Reviewer 1 comments

Reviewer #1: The authors theoretically demonstrated a class of plasmonic coupled elliptical nanostructure for achieving a spaser or a nanolaser with high intensity. Technically, the authors employ the numerical methods with apparent authority and the results seem valid. This work is interesting for the readership and deserve publication after major revision. 

Author Reply: We thank the respected Reviewer for appreciating our work and accepting our efforts. 

Concern 1. Figs. 2-9 (c) and (d) are blurred. Please replace them as the clear ones.

Author Reply: We thank the respected Reviewer for this comment and for highlighting the blurred images. The Figs.2-9 (c) and (d) have been modified with the high pixel format in the revised version of the manuscript. 

Concern 2. The references used in introduction section should be improved. It is written in abstract section that “Giant localized field enhancement and high LSPR values enable the proposed model to be highly appealing for sensing applications, surface enhanced Raman spectroscopy”. To be beneficial for the readers to enrich the mechanism and background of plasmonic sensor and surface enhanced Raman spectroscopy, the authors should mention the other approaches of plasmonic sensor, i.e., Nanomaterials, 10, 1399 (2021), J Phys D: Appl Phys, 54, 115301(2021)), Results in Physics, 15, 102567(2019) and Nanomaterials, 9,1691 (2019), and for surface enhanced Raman spectroscopy (i.e., Optics Express, 21, 24460 (2013)).

Author Reply: We are greatly thankful to the respected Reviewer for highlighting the weaker portion of the manuscript and for suggesting valuable related referenced papers. The suggested papers were worth reading and results in the enhancement of our knowledge. The suggested references were valid too and has been added in the revised version of the manuscript. 

Thank you for your attention, valuable suggestions, and patience, and if you have any questions, please don't hesitate to contact me. 

Yours sincerely,

Corresponding authors*

Response to Reviewer 2 comments

Reviewer #2: The authors reported a class of plasmonic coupled elliptical nanostructure for achieving a spaser or a nanolaser with high intensity. The FEM calculations showed that the LSPR peaks and the local field intensity or near field enhancement (NFE) of the active nanosystem can be amplified to higher values by introducing symmetry-breaking techniques in the proposed ellipse and its variants. Giant localized field enhancement and high LSPR values enable the proposed model to be highly appealing for sensing applications, surface enhanced Raman spectroscopy, and much more. The research is significant, the amount of data is large, and the contrast for characteristic parameters of different-type nano-structures is clear. However, before possible publication, some of the listed points should be explained and revised for further improving the manuscript.

Concern 1. Authors theoretically investigated a class of plasmonic coupled elliptical nanostructure for achieving spaser, but the shape of SBNE structures were unusual and whether they can be prepared for practical application?

Author Reply: The nanoparticles can be fabricated with electron beam lithography. Scanning electron microscopy (SEM) can be used to verify the success of the lithography process and to identify the best fabrication parameters such as exposure dose and development time. Both the nanostructures for the SEM and those for the optical characterization can be fabricated on the same substrate by using exactly the same electron beam lithography procedure. As mentioned in the manuscript, we have performed simulations to investigate the optical properties of ellipse and associated variants by changes in geometry and incident light. Our work is theoretical and is based on Finite Element Method (FEM). The SBNE is taken from the main ellipse, this elliptical model provide points for confining SPs that leads for the formation of hotspots. On the same time this configuration has rounded corners due to which it can be fabricated using above mentioned methods and imprinting lithography, atomic force microscopy or the methods explained by [4,5].

Concern 2. Why the structure type with maximum NFE for Elliptical nano-structures for the xpolarization was SBLCNET, but the structure type with maximum NFE for Elliptical nano-structures for the y polarization was LCNEQ rather than SBLCNET?

Author Reply: Respected Reviewer, the high NFE for x-polarization was achieved by SBLCNET, due to more exposed area of nanostructures to light and plasmon confinement points specially on the tips. While, for y-polarization this configuration showed less NFE and high value achieved for LCNEQ for y-polaroid case because here the light direction was changed and all of the four ellipses were totally receiving incident light efficiently that led to maximum plasmon confinement in the structure and produced a high NFE value for this configuration.

Concern 3. The calculated results should be supported by some experimental data through experimental measurement or experimental results from literatures.

Author Reply: Respected Reviewer, the following table has been added in the revised version of the manuscript which shows a comparison of our work with others studies with both theoretical and experimental work.

Table A : Comparison of current study with other works

Ref. No Near Field Enhancement (NFE) No. of Peaks

6 300 1

7 50 3

8 80 (abundance %) 4

9 1.5 ev 3

10 90 1

11 14.8 1

12 143 3

Current study 1019 5

Concern 4. How does the performances (such as NFE) of the structures in this work when compared with that in literatures.

Author Reply: Table A in the revised version of the manuscript summarizes the performance of current study with the others study in terms of NFE and LSPR/extinction/scattering peaks. 

Concern 5. There were too many abbreviations, so it is recommended to put them in a table.

Author Reply: We thank the respected Reviewer for suggesting abbreviation table. The point is valid, and table has been added in the revised version of the manuscript. 

Concern 6. The table note for Table 3 should be Table 2.

Author Reply: We thank the respected Reviewer for highlighting this error. The correct note for tables has been performed in the revised version of the manuscript. 

Thank you for your attention, valuable suggestions, and patience, and if you have any questions, please don't hesitate to contact me. 

Yours sincerely,

Corresponding authors*

 References:

[1]. Kazanskiy, N.L., Khonina, S.N., Butt, M.A., Kaźmierczak, A. and Piramidowicz, R., 2021. A numerical investigation of a plasmonic sensor based on a metal-insulator-metal waveguide for simultaneous detection of biological analytes and ambient temperature. Nanomaterials, 11(10), p.2551.

[2]. Chao, C.T.C., Chau, Y.F.C. and Chiang, H.P., 2021. Highly sensitive metal-insulator-metal plasmonic refractive index sensor with a centrally coupled nanoring containing defects. Journal of Physics D: Applied Physics, 54(11), p.115301.

[3]. Chau, Y.F.C., Chao, C.T.C., Huang, H.J., Anwar, U., Lim, C.M., Voo, N.Y., Mahadi, A.H., Kumara, N.T.R.N. and Chiang, H.P., 2019. Plasmonic perfect absorber based on metal nanorod arrays connected with veins. Results in Physics, 15, p.102567.

[4]. Chou Chau, Y.F., Chen, K.H., Chiang, H.P., Lim, C.M., Huang, H.J., Lai, C.H. and Kumara, N.T.R.N., 2019. Fabrication and characterization of a metallic–dielectric nanorod array by nanosphere lithography for plasmonic sensing application. Nanomaterials, 9(12), p.1691.

[5]. Tseng, M.L., Chang, C.M., Cheng, B.H., Wu, P.C., Chung, K.S., Hsiao, M.K., Huang, H.W., Huang, D.W., Chiang, H.P., Leung, P.T. and Tsai, D.P., 2013. Multi-level surface enhanced Raman scattering using AgO x thin film. Optics express, 21(21), pp.24460-24467.

[6]. Haynes, C.L., McFarland, A.D. and Van Duyne, R.P., 2005. Surface-enhanced Raman spectroscopy.

[7]. Kelly, K.L., Coronado, E., Zhao, L.L. and Schatz, G.C., 2003. The optical properties of metal nanoparticles: the influence of size, shape, and dielectric environment. The Journal of Physical Chemistry B, 107(3), pp.668-677.

[8]. Chen, S.; Carroll, D. Nano Lett. 2002, 2, 1003–1007

[9]. T.W. Ebbesen, H.J. Lezec, H. Ghaemi, T. Thio, P.A. Wolff, Extraordinary optical transmission through sub-wavelength hole arrays. Nature 391, 667–669 (1998).

[10]. Huo, Y.Y., Jia, T.Q., Zhang, Y., Zhao, H., Zhang, S.A., Feng, D.H. and Sun, Z.R., 2014. Spaser based on Fano resonance in a rod and concentric square ring-disk nanostructure. Applied Physics Letters, 104(11), p.113104.

[11]. Zhang, H., Zhou, J., Zou, W. and He, M., 2012. Surface plasmon amplification characteristics of an active three-layer nanoshell-based spaser. Journal of Applied Physics, 112(7), p.074309.

[12]. Huo, Y., Jia, T., Zhang, Y., Zhao, H., Zhang, S., Feng, D. and Sun, Z., 2013. Narrow and deep Fano resonances in a rod and concentric square ring-disk nanostructures. Sensors, 13(9), pp.11350-11361.

---

## [Decision Letter · Decision Letter 1]

24 Jan 2022

Large Electromagnetic Field Enhancement in Plasmonic Nanoellipse for Tunable Spaser Based Applications

PONE-D-21-35833R1

Dear Dr. Farooq,

We’re pleased to inform you that your manuscript has been judged scientifically suitable for publication and will be formally accepted for publication once it meets all outstanding technical requirements.

Kind regards,

Yuan-Fong Chou Chau

Academic Editor

PLOS ONE

Additional Editor Comments (optional):

Reviewers' comments:

Reviewer's Responses to Questions

**Comments to the Author**

1. If the authors have adequately addressed your comments raised in a previous round of review and you feel that this manuscript is now acceptable for publication, you may indicate that here to bypass the “Comments to the Author” section, enter your conflict of interest statement in the “Confidential to Editor” section, and submit your "Accept" recommendation.

Reviewer #1: All comments have been addressed

2. Is the manuscript technically sound, and do the data support the conclusions?

Reviewer #1: Yes

3. Has the statistical analysis been performed appropriately and rigorously? 

Reviewer #1: Yes

4. Have the authors made all data underlying the findings in their manuscript fully available?

Reviewer #1: Yes

5. Is the manuscript presented in an intelligible fashion and written in standard English?

Reviewer #1: Yes

6. Review Comments to the Author

Reviewer #1: The authors have revised their mauscript according to my comments. This manuscript can now be accepted for publication.

7. PLOS authors have the option to publish the peer review history of their article (what does this mean?). If published, this will include your full peer review and any attached files.

Reviewer #1: No

---

## [Editor Report · Acceptance letter]

8 Mar 2022

PONE-D-21-35833R1 

Large Electromagnetic Field Enhancement in Plasmonic Nanoellipse for Tunable Spaser Based Applications 

Dear Dr. Farooq:

I'm pleased to inform you that your manuscript has been deemed suitable for publication in PLOS ONE. Congratulations! Your manuscript is now with our production department. 

Kind regards, 

on behalf of

Dr. Yuan-Fong Chou Chau 

Academic Editor

PLOS ONE